# Detecting latent interaction effects when analyzing binary traits

Ziang Zhang[1,5], Jerald F. Lawless[2], Andrew D. Paterson[3,4*], Lei Sun[3,5*]

**1** Department of Human Genetics, University of Chicago, Chicago, Illinois, United States of America,
**2** Department of Statistics and Actuarial Science, University of Waterloo, Waterloo, Ontario, Canada,
**3** Division of Biostatistics, Dalla Lana School of Public Health, University of Toronto, Toronto, Ontario,
Canada, **4** Program in Genetics & Genomic Biology, The Hospital for Sick Children Research Institute,
Toronto, Ontario, Canada, **5** Department of Statistical Sciences, University of Toronto, Toronto, Ontario,
Canada

* andrew.paterson@sickkids.ca; lei.sun@utoronto.ca

UNITED KINGDOM OF GREAT BRITAIN AND
NORTHERN IRELAND

**Peer Review History:** PLOS recognizes the
benefits of transparency in the peer review
process; therefore, we enable the publication of
all of the content of peer review and author
responses alongside final, published articles.
The editorial history of this article is available
here: https://doi.org/10.1371/journal.pgen.
1011822

**Data availability statement:** This research has
been conducted using the UK Biobank Resource
under Application Number 64875. Data are

## Abstract

In genome-wide association studies (GWAS), it is often desirable to test for interactions, such as gene–environment ($G \times E$) or gene–gene ($G \times G$) interactions, between single-nucleotide polymorphisms (SNPs, $G$'s) and environmental variables ($E$'s). However, directly accounting for interaction is often infeasible, because the interacting variable is latent or the computational burden is too large. For quantitative traits ($Y$) that are approximately normally distributed, it has been shown that indirect testing on $GxE$ can be done by testing for heteroskedasticity of $Y$ between genotypes. However, when traits are binary, the existing methodology based on testing the heteroskedasticity of the trait across genotypes cannot be generalized. In this paper, we propose an approach to indirectly test interaction effects for binary traits and subsequently propose a joint test that accounts for the main and interaction effects of each SNP during GWAS. The final method is straightforward to implement in practice—it simply involves adding a non-additive (i.e., dominance) term to standard GWAS additive models for binary traits and testing its significance. We illustrate the statistical features including type-I-error control and power of the proposed method through extensive numerical studies. Applying our method to the UK Biobank dataset, we showcase the practical utility of the proposed method, revealing SNPs and genes with strong potential for latent interaction effects.

## Introduction

It is well known that the interaction (denoted as $GxE$) between single-nucleotide polymorphisms (SNPs; $G$'s) and environmental factors ($E$'s), or between SNPs (denoted as $GxG$), play an important role in shaping human complex traits ($Y$'s) [14]. A classic $GxE$ example is the interaction effect between genetic variants in *PAH* and diet on the risk of phenylketonuria and its subsequent intellectual disability [11]. Examples of $GxG$ have also been reported by [21].

However, a direct, exhaustive $GxG$ search may be undesirable in the genome-wide association study (GWAS) setting because of the large-scale multiple hypothesis testing and the substantial computational burden involved. A direct $GxE$ analysis, on the other hand, may

available at https://www.ukbiobank.ac.uk with the permission of UK Biobank. The code used to replicate the analysis results in this paper can be accessed at github.com/AgueroZZ/GE_code_repo. The ethics approval of UK Biobank has been obtained from the North West Multi-centre Research Ethics Committee (MREC). The GWAS summary statistics described in Sect 5 are publicly available at https://zenodo.org/records/16279185.

**Funding:** This research was funded by the Natural Sciences and Engineering Research Council of Canada (NSERC; url: https://www.nserc-crsng.gc.ca/index_eng.asp; Grant Number: RGPIN-04934 to LS), the Center for Addiction and Mental Health Discovery Fund Seed Funding (CAMH; url: https://www.camh.ca/en/science-and-research/discovery-fund-seed-funding-projects; to LS), the University of Toronto Data Sciences Institute Catalyst Grant (DSI; url: https://datasciences.utoronto.ca/; to LS), and the Canadian Statistical Sciences Institute (CANSSI-STAGE; url: https://stage.utoronto.ca/; to ZZ). The funders had no role in study design, data collection and analysis, decision to publish, or preparation of the manuscript.

**Competing interests:** The authors have declared that no competing interests exist.

be infeasible in practice if the interacting $E$ is latent or missing. For these reasons, it is often useful to conduct latent interaction analyses in GWAS; to simplify the notations, we use $GxE$ hereinafter for both $GxE$ and $GxG$ scenarios.

For a quantitative trait $Y$ that is approximately normally distributed, it has been shown that a latent $E$ (or an un-modeled genetic variant) that interacts with a bi-allelic SNP $G$ will produce heteroskedasticity in $Y$ across the three genotypes of the SNP [20]. Consequently, SNPs for which traits have shown significant heteroskedasticity (variance-quantitative trait loci, vQTLs) can be used to screen for potential $GxE$, and multiple vQTL methods have been developed [15,16,22,23,25,26]. The vQTL latent interaction approach has identified promising SNPs for follow-up interaction analysis. For example, rs12753193 near *LEPR* was first identified through vQTL analysis with evidence of interaction effect with BMI on C-reactive protein levels [20].

The lack of a corresponding latent $GxE$ method for binary traits causes us to potentially miss novel findings. However, the variance technique for a quantitative trait cannot be used for a binary $Y$, as the variance of a binary trait is determined by its mean. Similarly, an over-dispersion parameter cannot be identified when $Y$ is binary [9]. Thus, how to indirectly detect latent $GxE$ in the binary setting remains an open problem.

In this paper, we first show that for binary traits commonly analyzed through the logistic and probit regression models, the latent $GxE$ can be indirectly tested as a non-additive effect of $G$. Analogous to the joint location-scale test for a quantitative trait that integrates the vQTL information with the traditional mean-based GWAS [22], we then show how the joint test for a binary trait is related to the so-called genotypic test. As a result, the proposed method can be smoothly integrated into existing GWAS pipelines such as PLINK2 [3] for binary traits by simply including a non-additive term and testing its effect. Finally, we demonstrate the validity, power and practical applicability of the proposed method through extensive numerical studies and real data application.

## Preliminary

**Indirect Test of Latent Interactions for Quantitative Traits.**

Let $Y$ be the trait of interest and $G$ the genotypes of the SNP of interest, with the major and minor alleles coded as $a$ and $A$, respectively, and the corresponding allele frequencies of $q = 1-p$ and $p$ ($\leq 0.5$), respectively; $p$ is the minor allele frequency (MAF). Furthermore, let $G_A$ denote the count of minor alleles $A$ at a SNP, then $G_A = 0, 1$ and $2$ corresponds to $G = aa, Aa$ and $AA$, respectively, also termed additive coding.

Although the proposed method applies when the interacting variable is either genetic (e.g., $GxG$) or environmental (e.g., $GxE$), we assume for now that the interacting variable is environmental and denote it by $E$, without loss of generality. If the environmental variable $E$ is observed and hypothesized to interact with $G$, then the following linear regression would typically be used:

$$Y = \beta_0 + \beta_G G_A + \beta_E E + \beta_{GE} G_A E + e, \tag{1}$$

where $e \sim N(0, \sigma_e^2)$ is independent of $G$ and $E$, and $G$ is typically assumed to be independent of $E \sim N(0, \sigma_E^2)$ [20]. In practice, the model often includes other covariates, which are omitted here from notation for simplicity but without loss of generality [23].

In many GWAS, the interacting $E$ may not be measured. Consequently, the working model will be

$$Y = \beta_0 + \beta_G G_A + e_G, \tag{2}$$

where both $E$ and its interaction $G_A E$ in [1] are latent.

This misspecified working model leads to heteroskedasticity which can be leveraged to indirectly test for the latent interaction. More specifically, the *variance* of the new random error $e_G$ has the following form,

$$\text{Var}(e_G|G_A) = \text{Var}(Y|G_A) = (\beta_E + \beta_{GE}G_A)^2\sigma_E^2 + \sigma_e^2, \tag{3}$$

which depends on $G_A$ if $\beta_{GE} \neq 0$. For this reason, various vQTL methods based on Levene-type tests [20,23] or quantile regression method [16] have been proposed to identify latent interactions and prioritize genetic variants for follow-up analyses.

**vQTL Approach Does Not Work for Binary Traits.**

To indirectly test if $\beta_{GE} = 0$ for binary traits, it might seem intuitive to extend the vQTL approach used for quantitative traits. However, the vQTL framework is not applicable to binary traits due to a fundamental difference: unlike a quantitative trait, the variance of a binary trait is inherently determined by its mean,

$$\text{Var}(Y|G_A) = \mathbb{E}(Y|G_A)[1 - \mathbb{E}(Y|G_A)]. \tag{4}$$

Thus, $\text{Var}(Y|G_A)$ does not yield additional information pertinent to the latent interaction effect. The over-dispersion approach, unfortunately, is not applicable either, as the over-dispersion parameter cannot be identified when $Y$ is binary [9].

**Regression Models for Binary Traits.**

When the trait of interest $Y$ is binary, the standard linear association model [1] is replaced by the following generalized linear model (GLM) [18],

$$\mathbb{P}(Y = 1|G_A, E) = g^{-1}(\beta_0 + \beta_G G_A + \beta_E E + \beta_{GE}G_A E), \tag{5}$$

where $g^{-1}$ refers to the inverse of the GLM link function. Depending on whether a logistic or probit regression is used, $g^{-1}$ corresponds to the CDF of the standard logistic or normal distribution, respectively.

An equivalent parametrization of model [5] above is through the latent regression formulation [5],

$$Y = \mathbb{I}\{Y^* \geq 0\}, \quad Y^* = \beta_0 + \beta_G G_A + \beta_E E + \beta_{GE}G_A E + \epsilon, \tag{6}$$

where $Y^*$ is latent; the importance of this latent model formulation will be apparent later. Similar to model [1], $G$ and $E$ are assumed to be independent of each other, and $\epsilon$ is assumed to be independent of $G$ and $E$. But, the error term $\epsilon$ here has a known distribution that is symmetric around zero with CDF $F_\epsilon$, whereas the error term $e$ in model [1] typically assumes a normal distribution with an unknown variance $\sigma_e^2$. For example, $\epsilon$ can follow either the standard logistic distribution or the standard normal distribution, corresponding to a logistic or probit regression model through the GLM formulation in [5].

Given the observed values of $G_A$ and $E$, the conditional probability of being a case (i.e. $Y = 1$) is,

$$\begin{aligned}
\mathbb{P}(Y = 1|G_A, E) &= \mathbb{E}\big[\mathbb{I}(Y^* \geq 0)|G_A, E\big] \\
&= \mathbb{P}\big(\epsilon \geq -(\beta_0 + \beta_G G_A + \beta_E E + \beta_{GE}G_A E)\big) \\
&= \mathbb{P}\big(\epsilon \leq \beta_0 + \beta_G G_A + \beta_E E + \beta_{GE}G_A E\big) \\
&= F_\epsilon(\beta_0 + \beta_G G_A + \beta_E E + \beta_{GE}G_A E).
\end{aligned} \tag{7}$$

The consequence of missing $E$ and its interaction in this model will be examined in greater detail in the next section, followed by the proposed method to detect the latent interaction.

## Methods

### Ethics Statement.

This research has been conducted using the UK Biobank Resource under Application Number 64875. The ethics approval of UK Biobank has been obtained from the North West Multi-centre Research Ethics Committee (MREC).

### Latent Interaction Test Based on Non-Additive Effect for Binary Traits.

Assume now the environmental variable $E$, thus also $G_A E$, in model [7] is latent, the probability of being a case is now

$$
\begin{aligned}
\mathbb{P}\big[Y=1|G_A\big] &= \mathbb{E}\big[\mathbb{I}(Y^* \geq 0)|G_A\big] \\
&= \mathbb{P}\big[\epsilon - (\beta_E + \beta_{GE}G_A)E \leq \beta_0 + \beta_G G_A|G_A\big] \\
&= \mathbb{P}\big[\epsilon - E^* \leq \beta_0 + \beta_G G_A|G_A\big],
\end{aligned}
\tag{8}
$$

where $E^* = (\beta_E + \beta_{GE}G_A)E$. It is then obvious that $G_A$ and $\epsilon - E^*$ are independent of each other if and only if $\beta_{GE} = 0$.

To simplify the presentation, we make Assumption 1 without the loss of generality.

**Assumption 1.** *The conditional distribution of $\epsilon - E^*$ given $G_A$ is in a certain location-scale family. So $\epsilon^* = (\epsilon - E^*)/SD(\epsilon - E^*|G_A)$ has a completely specified CDF $F_{\epsilon^*}$ that does not depend on $G_A$.*

**Remark 1.** *When $E \sim N(0, \sigma_E^2)$, Assumption 1 will often not hold unless $F_\epsilon$ is the standard normal CDF. However, for commonly used models such as logistic regression, Assumption 1 holds approximately due to the close relationship between the standard normal and logistic distributions [2]. Assumption 1 is only used to simplify the presentation in the rest of this paper; the proposed method remains valid without this assumption.*

If $\beta_{GE} = 0$ and we define $c = SD(\epsilon - E^*)$, Assumption 1 implies that $\epsilon^* = (\epsilon - E^*)/c$ has a completely specified distribution $F_{\epsilon^*}$, and hence Eq (8) becomes:

$$
\mathbb{P}\big(Y=1|G_A\big) = F_{\epsilon^*}\left(\frac{\beta_0 + \beta_G G_A}{c}\right).
\tag{9}
$$

In other words, fitting a binary model with link function $F_{\epsilon^*}$ can correctly recover all regression coefficients up to a positive scaling, hence the testing of $\beta_G = 0$ is not affected.

When $\beta_{GE} \neq 0$, the variable $\epsilon - E^*$ will depend on $G_A$ through its standard deviation (SD):

$$
c(G_A) = SD(\epsilon^*|G_A) = \sqrt{\sigma_\epsilon^2 + (\beta_E + \beta_{GE}G_A)^2\sigma_E^2},
\tag{10}
$$

which implies:

$$
\mathbb{P}\big(Y=1|G_A\big) = F_{\epsilon^*}\left(\frac{\beta_0 + \beta_G G_A}{c(G_A)}\right).
\tag{11}
$$

This model is no longer linear on $G_A$, but since $G_A$ only has values of 0, 1 or 2, model [11] is saturated. This implies model [11] can always be fully parameterized with three parameters

without the problem of model-misspecification. In particular, model [11] can be rewritten as:

$$\mathbb{P}(Y = 1|G) = F_{\epsilon*}\left(\gamma_0 \mathbb{I}\{G = aa\} + \gamma_1 \mathbb{I}\{G = Aa\} + \gamma_2 \mathbb{I}\{G = AA\}\right), \qquad (12)$$

with the parameters $\gamma_0, \gamma_1$ and $\gamma_2$ defined as

$$
\begin{aligned}
\gamma_0 &= \frac{\beta_0}{c(G_A = 0)} = \frac{\beta_0}{\sqrt{(\beta_E^2 \sigma_E^2 + \sigma_\epsilon^2)}}, \\
\gamma_1 &= \frac{\beta_0 + \beta_G}{c(G_A = 1)} = \frac{\beta_0 + \beta_G}{\sqrt{((\beta_E + \beta_{GE})^2 \sigma_E^2 + \sigma_\epsilon^2)}}, \\
\gamma_2 &= \frac{\beta_0 + 2\beta_G}{c(G_A = 2)} = \frac{\beta_0 + 2\beta_G}{\sqrt{((\beta_E + 2\beta_{GE})^2 \sigma_E^2 + \sigma_\epsilon^2)}},
\end{aligned}
\qquad (13)
$$

where $\sigma_\epsilon$ is $\pi/\sqrt{3}$ for logistic regression and 1 for probit regression.

**Remark 2.** *In practice, since it is difficult to explicitly know the distribution of $\epsilon^*$ and hence to fit the corresponding binary model [12], it is easier to fit the binary model with the original link function $F_\epsilon$. Since Eq (12) is a saturated model, using a different link function will not introduce any problem of model-inadequacy.*

Define the non-additive effect as:

$$\gamma_D = (\gamma_2 - \gamma_1) - (\gamma_1 - \gamma_0) = \gamma_2 - 2\gamma_1 + \gamma_0. \qquad (14)$$

It is clear that if $\beta_{GE} = 0$, the working model [12] is additive with $\gamma_D = 0$. A non-additive effect $\gamma_D$ is created in the model [12] for $Y$ given $G$ when there is a latent interaction in the model [7] for $Y$ given $G$ and $E$. Therefore, analogous to using vQTLs to detect *GxE* effect in the analysis of quantitative traits, the *GxE* effect in the analysis of binary traits can be indirectly detected from testing the non-additive effect $\gamma_D$, which is elsewhere termed the dominance effect.

**Equivalent Parameterization of Genotypic Models.**

From Eqs (11) and (12), it can be noticed that the working model can always be written as a binary regression model with the genotypic encoding in Eq (12). Since the model is saturated, there are many equivalent re-parametrizations of this genotypic model with three regression parameters. To assess how much non-additive genetic variation is created by the latent *GxE* compared to the additive genetic variation, we consider the Fisher orthogonal re-parametrization of this model under the Hardy-Weinberg equilibrium (HWE) assumption of the SNP:

$$
\begin{aligned}
\mathbb{P}(Y = 1|G_A) &= F_{\epsilon*}\left(\gamma_0 \mathbb{I}\{G_A = 0\} + \gamma_1 \mathbb{I}\{G_A = 1\} + \gamma_2 \mathbb{I}\{G_A = 2\}\right), \\
&= F_{\epsilon*}\left(\beta_0^* + \beta_A^* G_A + \beta_D^* G_D\right),
\end{aligned}
\qquad (15)
$$

where $G_A = (0, 1, 2)$ and $G_D = (-p/q, 1, -q/p)$ for the genotypes $(aa, Aa, AA)$. Given a vector of three genotypic effects $\gamma = (\gamma_0, \gamma_1, \gamma_2)^T$ for the genotypes $(aa, Aa, AA)$, the parameters $\beta_A^*$ and

$\beta_D^*$ in Eq (15) can be computed as:

$$\beta_A^* = \boldsymbol{L}_A \boldsymbol{\gamma} = p\gamma_D + (\gamma_1 - \gamma_0), \quad \boldsymbol{L}_A = [-q, q-p, p],$$
$$\beta_D^* = \boldsymbol{L}_D \boldsymbol{\gamma} = -pq\gamma_D, \quad \boldsymbol{L}_D = [-pq, 2pq, -pq]. \tag{16}$$

S1 Table summarizes the two equivalent parameterizations of Equation 12, along with the expressions for their regression coefficients in terms of the original parameters from the true model.

This Fisher orthogonal encoding ensures that the two variables $G_A$ and $G_D$ are uncorrelated [8,19], and therefore the proportion of genetic effect explained by the non-additive component (on the latent $Y^*$) is

$$R_D^2 = \frac{\mathrm{Var}(\beta_D^* G_D)}{\mathrm{Var}(\beta_A^* G_A + \beta_D^* G_D)} = \frac{\beta_D^{*2}}{2pq\beta_A^{*2} + \beta_D^{*2}}. \tag{17}$$

To illustrate the non-additive effect $\gamma_D$ introduced by the latent *GxE*, we show the values of $\gamma_D$ under different parameter settings in Eq (5), for a probit regression model, in Fig 1(a-b) and Supplementary S1 Fig (a-b). Although the original model [5] only contains the additive component $G_A$, if there exists a latent interaction (i.e. $\beta_{GE} \neq 0$), a non-negligible $\gamma_D$ is induced for most of the parameter settings. The non-additive effect $\gamma_D$ represents the deviation from linearity in the genotype-dependent component of the linear predictor, specifically capturing the extent to which the value for the heterozygous genotype ($G = 1$) deviates from the mid-point between the two homozygous genotypes ($G = 0$ and $G = 2$). A negative $\gamma_D$ indicates a form of sub-additivity, where the effect of carrying two copies of the risk allele is disproportionately smaller than expected under an additive model. Notably, such a pattern can emerge even when both $\beta_G$ and $\beta_{GE}$ are positive, due to the non-linear composition of functions in Eq (11) that define the binary outcome model (as illustrated in Fig 1(a-b)). Note that Fig 1(b) is simply a mirror image of Fig 1(a) with respect to $\beta_{GE}$, because $\gamma_D$ remains unchanged when both $\beta_E$ and $\beta_{GE}$ are replaced by $-\beta_E$ and $-\beta_{GE}$, as shown in Eq (13).

In Fig 1(c-d) and Supplementary S1 Fig (c-d), we compute the corresponding $R_D^2$ as defined in Eq (17) for the same sets of parameters. In most settings, the induced non-additive component comprises a moderate proportion of the genetic variation; however, when the minor allele of the SNP has a protective effect ($\beta_G < 0$) for the binary trait, the non-additive proportion is particularly large. Similar to $\gamma_D$, the proportion of variance explained by the non-additive component depends in a highly non-linear manner on all of the regression parameters, due to the complex functional composition in Eq (11) and the non-linear link function.

**Wald Test.**

The choice of test does not affect the validity of the proposed method, and given the large sample sizes typical in GWAS settings, these tests are expected to have comparable power. The Wald test is often used in GWAS practice because fitting the model to perform the test also yields maximum likelihood estimates and standard errors for each covariate, enabling straightforward reporting of effect sizes and p-values. Therefore, we adopt the Wald test to maintain consistency with existing literature [27] and widely used software implementations such as PLINK2 [3]. We note, however, that the method presented in this work is also compatible with other types of tests. For example, when only a p-value is needed from a model without additional covariates, a score test can be used to improve computational efficiency.

Although GWAS regression models, including our example in Section 5, typically include auxiliary covariates such as age, sex, and genetic PCs, in this section we consider the

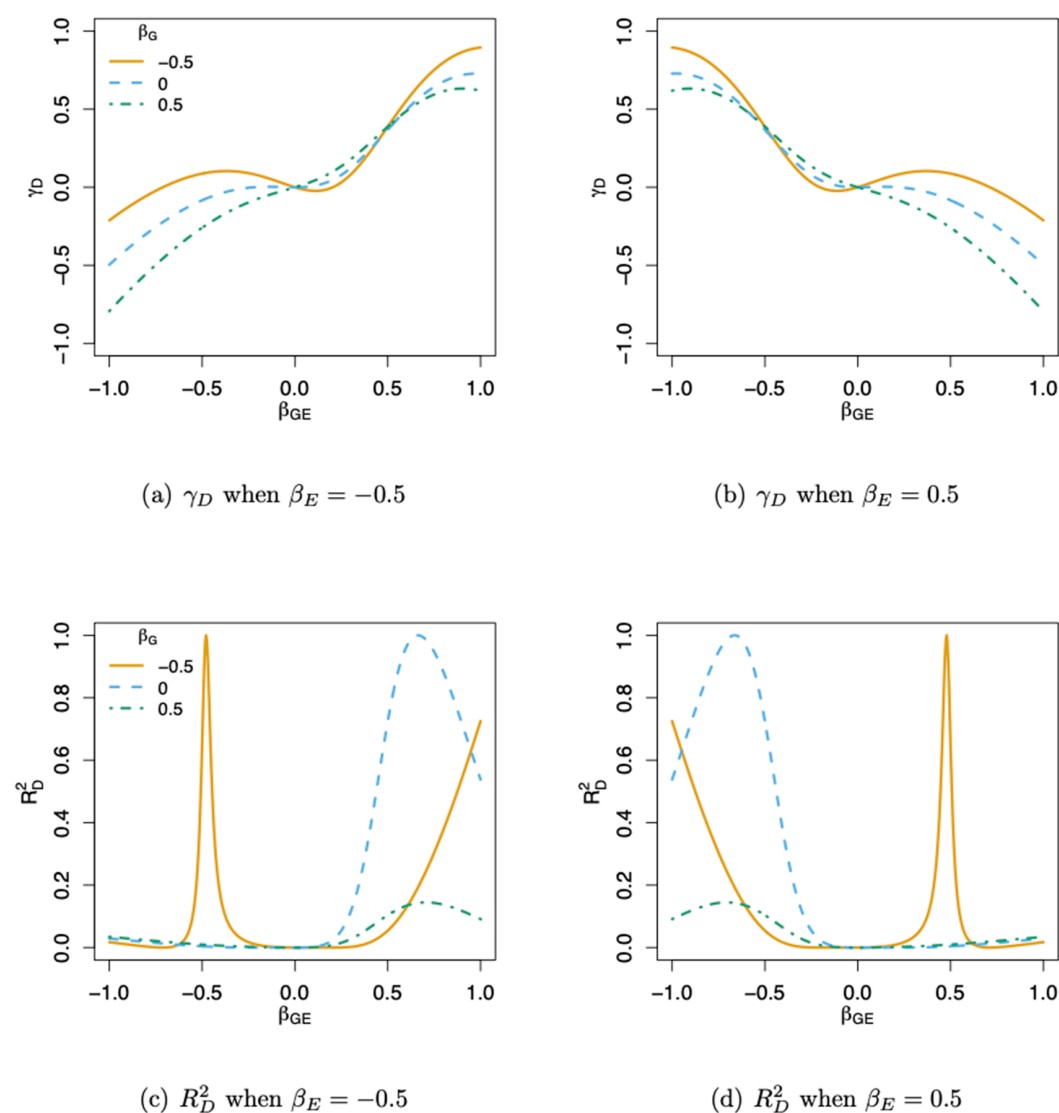

(a) $\gamma_D$ when $\beta_E = -0.5$ (b) $\gamma_D$ when $\beta_E = 0.5$

(c) $R_D^2$ when $\beta_E = -0.5$ (d) $R_D^2$ when $\beta_E = 0.5$

**Fig 1.   (a-b) show values of $\gamma_D$ and (c-d) show the non-additive proportion of genetic variation $R_D^2$, at different $\beta_G$ and $\beta_{GE}$.** The covariate $E \sim N(0,1)$ is assumed to have a main effect of $\beta_E = -0.5$ (left column) and 0.5 (right column). The MAF is set to $p = 0.3$ in (c-d). The underlying model is assumed to be probit. The prevalence of the binary trait $Y$ is assumed to be 0.1.

simplified model [15] without covariates, which suffices to illustrate the key ideas and does not affect the generality of our conclusions. Let $\widehat{\beta^*}$ denotes the maximum likelihood estimate (MLE) of the vector of regression parameter $\beta^* = (\beta_0^*, \beta_A^*, \beta_D^*)^T$ in model [15], $L \in \mathbb{R}^{d \times 3}$ denotes the constraint matrix with $d$ linear independent rows for the null hypothesis $H_0 : L\beta^* = \mathbf{0}$. The Wald test uses the test statistics:

$$T = (L\widehat{\beta^*})^T (L I_n^{-1}(\widehat{\beta^*}) L^T)^{-1} (L\widehat{\beta^*}) \tag{18}$$

where $I_n(\widehat{\beta^*})$ denotes the Fisher information matrix evaluated at the MLE. Under the null hypothesis, the test statistics $T$ asymptotically follows a chi-square distribution with $d$ degrees

of freedom as the sample size $n$ grows. To indirectly detect the latent interaction effect $\beta_{GE}$, we test the non-additive effect $\beta_D^*$ of the SNP, which corresponds to $L = [0, 0, 1] \in \mathbb{R}^{1 \times 3}$. Since $\beta_D^* = pq\gamma_D = 0$ whenever $\beta_{GE} = 0$, the proposed non-additive test will have the correct test size.

In traditional GWAS, the testing of SNPs is typically based on their additive main effects, while omitting possible non-additive effects [19]. This additive-only approach corresponds to $L = [0, 1, 0] \in \mathbb{R}^{1 \times 3}$ in the Wald test. Since the latent $GxE$ for binary trait ($\beta_{GE}$) induces a non-additive genetic effect $\gamma_D$, or equivalently, $\beta_D^*$ in the working model, we propose a joint test of the hypothesis $H_0 : \beta_A^* = \beta_D^* = 0$ in model [15], in order to detect the latent interaction $\beta_{GE}$ together with the main effect $\beta_G$ in Eq (5). This uses a constraint matrix $L \in \mathbb{R}^{2 \times 3}$ that specifies the null hypothesis $\beta_D^* = \beta_A^* = 0$, and the Wald test has two degrees of freedom, in contrast to the one degree of freedom test that only considers the additive effect $\beta_A^*$. We emphasize that the proposed joint test is not restricted to the Fisher orthogonal encoding in Eq (15). In fact, the two degrees of freedom joint test can be equivalently performed using any saturated model that encodes the genotypic effects with three regression parameters. The encoding in Eq (15) is used to simplify the partition of additive and non-additive effects.

## Method verification

### Type I Error Evaluation.

In this section, we will assess the type I error rate of the proposed non-additive test and the joint test, respectively for testing $\beta_{GE} = 0$ and $\beta_G = \beta_{GE} = 0$.

To assess the type I error rate of the proposed non-additive test that detects the latent $GxE$ effect $\beta_{GE}$ based on the non-additive effect $\beta_D^*$, we simulate $n = 100,000$ independent individuals under the *theoretical* null hypothesis that $\beta_{GE} = 0$, using the logistic model in Eq (6). We fix $\beta_0 = -1$ and $\beta_G = 0.5$, and independently simulate $E \sim N(0, 1)$ and $G$ under the HWE. We consider six settings defined by the combinations of minor allele frequency (MAF) of $G$ in $\{0.1, 0.3, 0.5\}$ and $\beta_E$ in $\{0, 1\}$. The prevalence of the trait varies roughly between 0.3 and 0.4 across these six settings. For each of the six settings, we perform $B = 100,000$ independent replications and compute one p-value per replication using the Wald test statistic in Eq (18), resulting in $B$ independent p-values. To also assess the type I error rate of the proposed joint test that accounts for the latent $GxE$ $\beta_{GE}$ together with the main effect $\beta_G$, we further fix the $\beta_G = 0$ and obtain the p-values of the proposed joint test from the same settings above. As shown in Table 1, the proposed non-additive and joint tests both have well-controlled type I error rates across different parameter settings. The histograms of the p-values of the

**Table 1. Empirical type I error rates of the proposed non-additive test (above) and the joint test (below) for each choice of $\beta_E$, MAF and significance level $\alpha$. The rates are computed using $B = 100,000$ independent replications, each with $n = 100,000$ simulated individuals.**

| $\beta_E$ | $\alpha = 0.05$ | | | $\alpha = 0.005$ | | | $\alpha = 0.0005$ | | |
|---|---|---|---|---|---|---|---|---|---|
| | MAF=0.1 | 0.3 | 0.5 | 0.1 | 0.3 | 0.5 | 0.1 | 0.3 | 0.5 |
| 0 | 0.0502 | 0.0509 | 0.0517 | 0.00485 | 0.00487 | 0.00512 | 0.00057 | 0.00050 | 0.00056 |
| 1 | 0.0503 | 0.0523 | 0.0520 | 0.00528 | 0.00568 | 0.00575 | 0.00048 | 0.00053 | 0.00050 |
| $\beta_E$ | $\alpha = 0.05$ | | | $\alpha = 0.005$ | | | $\alpha = 0.0005$ | | |
| | MAF=0.1 | 0.3 | 0.5 | 0.1 | 0.3 | 0.5 | 0.1 | 0.3 | 0.5 |
| 0 | 0.0506 | 0.0498 | 0.0509 | 0.00462 | 0.00541 | 0.00530 | 0.00040 | 0.00053 | 0.00060 |
| 1 | 0.0506 | 0.0504 | 0.0502 | 0.00520 | 0.00485 | 0.00488 | 0.00046 | 0.00052 | 0.00054 |

non-additive test and the joint test are provided in the supplement (S2 and S3 Figs), where the distributions of p-values are shown to be close to $\text{Unif}[0, 1]$ in all settings.

The p-values and the empirical type I error rates of the proposed non-additive and joint tests above are obtained under a *theoretical* null hypothesis, in which the traits were directly generated from a null model in which the hypothesis $\beta_{GE} = 0$ is true. As discussed in [28], another way to assess the test size of a method is through the *empirical* null hypothesis, in which the traits are generated from an alternative model, but are then randomly permuted before being tested. To further assess the type I error rate of the proposed tests under the empirical null hypothesis, we use data from the UK Biobank (UKB) [1] to implement a GWAS in a randomly permuted binary trait (self-reported) high cholesterol (Data-Field 20002; Coding 1473; Prevalence: 0.121), collected at the baseline. The details of the GWAS procedures are the same as those described later in the next section. The genomic-control (GC) $\lambda$ of the p-values of this permuted GWAS is computed to be 1.004 for the non-additive and 0.996 for the joint test [6]. The histograms and the QQ plots of these p-values are displayed in the supplementary material (S4 Fig).

**Power Comparison.**

In this section, we provide a detailed assessment of the powers of the proposed indirect test of the latent interaction $\beta_{GE}$, and the powers of the proposed joint test that simultaneously detects the main effect $\beta_G$ and the latent interaction effect $\beta_{GE}$. To simplify the power computation, we assume a probit model in Eq (5) as the true model, which satisfies Assumption 1. The SNP $G$ is generated with MAF = 0.3 under the assumption of HWE. The latent environmental variable $E$ follows $N(0, 1)$ with an effect $\beta_E = -0.5, 0$ and 0.5. The genetic effect $\beta_G$ and interaction effect $\beta_{GE}$ range from –1 to 1, and the sample size is set to $n = 30{,}000, 300{,}000$ and $800{,}000$. The intercept $\beta_0$ is set for a prevalence rate of 10 percent.

Since the probit regression model is assumed, the asymptotic power of the Wald test can be computed analytically for each case. First, we compute the corresponding values of $\beta_A^*$ and $\beta_D^*$ based on the values of $\beta_G$ and $\beta_{GE}$, using Eqs (13) and (16). Second, we compute the non-centrality parameter of the Wald test statistic $T$ as

$$\lambda = (L\beta^*)^T (L I_n^{-1}(\beta^*) L^T)^{-1} (L\beta^*).$$

Finally, the asymptotic power of the Wald test is computed using the non-centrality parameter $\lambda$ as:

$$1 - F_\lambda[F_0^{-1}(1 - \alpha)],$$

where $\alpha$ is set to the genome-wide significance level $5 \times 10^{-8}$ [7]; $F_\lambda$ denotes the CDF of the non-central Chi-square distribution with $d$ degrees of freedom and non-centrality parameter $\lambda$, and $F_0^{-1}$ denotes the inverse CDF of the central Chi-square distribution with $d$ degrees of freedom.

Fig 2 show the power of the proposed non-additive test of $\beta_{GE}$ and the proposed joint test of $\beta_G$ and $\beta_{GE}$, when the sample size $n = 300{,}000$ or $n = 30{,}000$. To aid interpretation of Fig 2, it may also be helpful to see how the size of the dominance effect $\gamma_D$ varies as a function of $\beta_G$, $\beta_{GE}$, $\beta_E$, and MAF. For this, we again refer the reader to the contour plots in S1 Fig. As shown in Fig 2, both the proposed non-additive and joint tests tend to have higher power when $\beta_{GE}$ and $\beta_E$ have opposite signs, which happens when the environmental variable has opposite effects dependent on the dosage of the minor allele of the SNP. In most cases, when either $\beta_G$ or $\beta_{GE}$ is away from 0, the proposed joint test has power close to 1 to detect the genetic signal. Yet, for certain values of $\beta_G$ and $\beta_{GE}$ that deviate significantly from 0, the joint test exhibits limited power to detect them. This occurs when the values of $\beta_G$ and $\beta_{GE}$ lead to both $\beta_D$ and

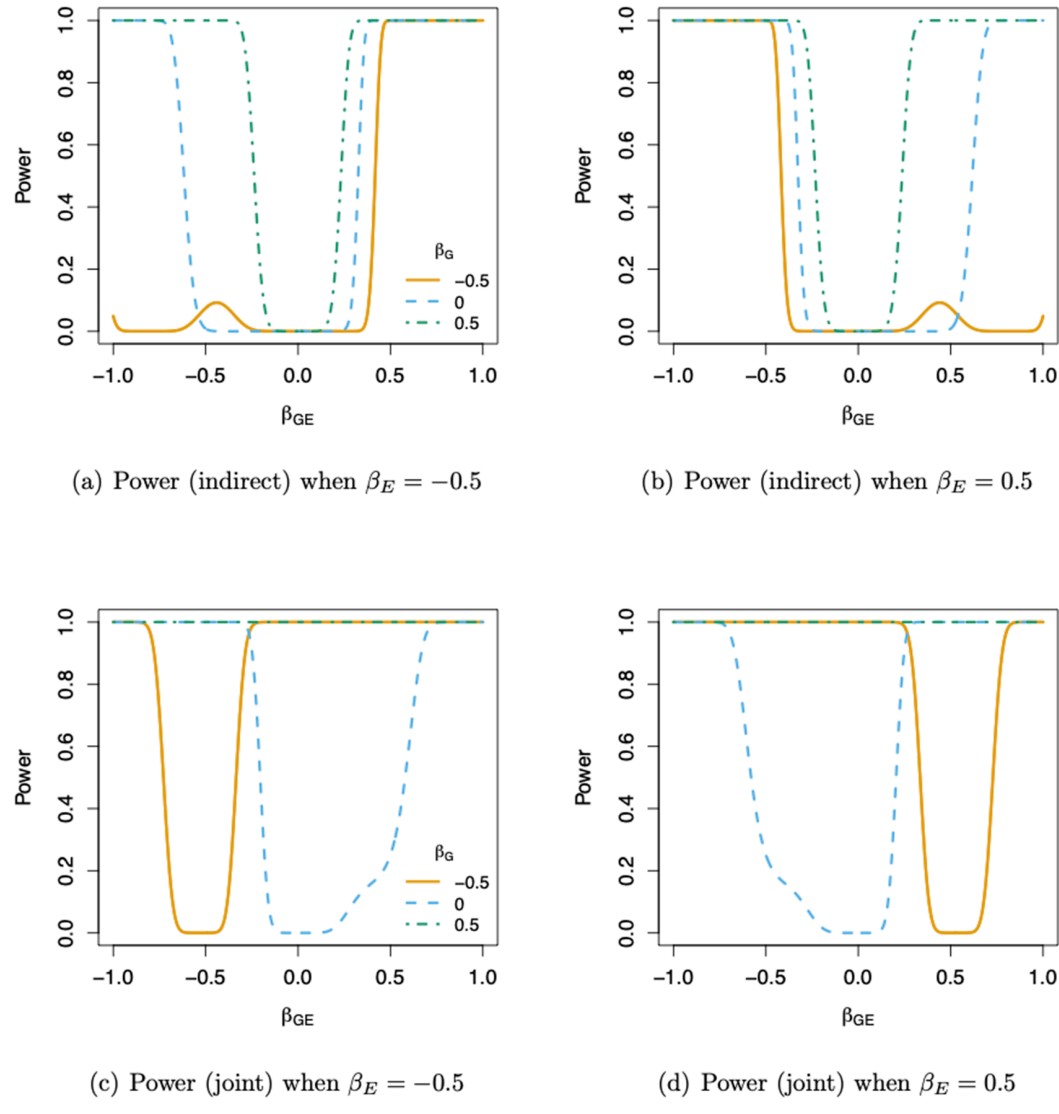

**Fig 2. Power of the proposed tests: The power for the proposed non-additive test (1 df) based on $\beta_D$ is shown in the first row, and the power for the proposed joint test (2 df) of $\beta_{GE}$ and $\beta_G$ is shown in the second row.** The sample sizes are set to $n = 300,000$ for the non-additive test and $n = 30,000$ for the joint test, in order to better illustrate their power behavior under different parameter settings. The significance level is set to $\alpha = 5 \times 10^{-8}$.

$\beta_A$ in Eq (15) being near zero. As illustrated in the supplement (S5 - S7 Figs), these instances become less frequent as the sample size $n$ increases.

## Results

We illustrated the usage of the proposed non-additive (indirect) test and its subsequent (2-df) joint test. We achieved this by a GWAS on UKB of the binary trait (self-reported) of high cholesterol (Data-Field 20002; Coding 1473) [1,24].

We selected genotyped SNPs with MAF greater than 0.01, HWE p-values greater than 1e-50 and SNP call rates greater than 0.8. This resulted in 626,164 autosomal SNPs being analyzed. To avoid the potential bias from ancestry, we restricted our analysis to unrelated self-reported British participants with ancestries further confirmed by the PC constructed from genetic data (Data-Field 22006). The related individuals were filtered out based on the kinship coefficients (Data-Field 22021), and we further filtered out individuals with genotype missing rates higher than 0.2. The final sample consists of 276,658 approximately unrelated individuals. The prevalence rate of the trait in the final sample is 0.121 (0.151 in males, 0.095 in females).

We then used logistic regression to analyze the genetic association between each SNP and the binary trait (high cholesterol), accounting for covariate effects of age (Data-Field 21022), sex (Data-Field 31) and first four principal components (PC) constructed from genetic data (Data-Field 22009). We carried out the GWAS using both the (2-df) joint test and the (1-df) non-additive test. The computation in this example was carried out using the software PLINK2 [3]. In PLINK2, the non-additive term $G_D$ is encoded using an over-dominance scheme (0, 1, 0 for genotypes *aa*, *Aa*, *AA*). Unlike the orthogonal encoding described earlier, this coding does not ensure that $G_A$ and $G_D$ are uncorrelated. While this non-orthogonality may affect the marginal test for $\beta_A$, it does not affect the results of the joint test ($\beta_A = \beta_D = 0$) or the test of the non-additive effect ($\beta_D = 0$).

The two GWAS results are displayed in Fig 3(a). As reflected in the Miami plot, we identified a number of SNPs with genome-wide significant association with high cholesterol using the joint test (GC $\lambda$ = 1.074). For completeness, the Manhattan plot from the traditional additive GWAS—where the dominance term $G_D$ is not included in the model—is also shown in Fig 3 (GC $\lambda$ = 1.139). Among these SNPs identified by the joint test, the non-additive test (GC $\lambda$ = 1.007) flagged 4 SNPs with genome-wide significant non-additive effects for follow-up studies of latent *GxE* effects, with the top rs7412 (p-value = 1.640e-19) in *APOE*. The QQ plots and histograms of the two GWAS can be found in the supplementary material (S8 Fig).

The latent interactions suggested by the non-additive test are not unexpected, given the well-established literature on the haplotype effects of *APOE* on cholesterol levels [4,17], which can be viewed as an interaction between nearby SNPs [12]. To investigate potential *GxG* interactions involving nearby SNPs, we selected 65 SNPs within 10,000 kb of rs7412 with $D'$ greater than 0.2, and performed pairwise *GxG* interaction analyses between these SNPs and rs7412. The LD information including $D'$, $r^2$ as well as the physical position of these SNPs were obtained using the tool LDlink [13], with the genome build GRCh37 and superpopulation of all the European groups (CEU, TSI, FIN, GBR and IBS). The histograms of p-values for the interaction tests and for the proposed indirect tests of the selected 65 SNPs are provided in Fig 4(a-b), and the full result can be found in S2 Table. Indeed, we found 11 SNPs having interaction with rs7412 at the significance level of 0.05 after Bonferroni correction; 5 of the 11 SNPs also have p-values less than 0.05 using the proposed non-additive test based on the non-additive effect. The SNP with the smallest p-value of the interaction test (1.713e-07) is rs7254892, which is mapped to *NECTIN2*. This SNP has $D'$ = 1 with rs429358, which in combination with rs7412 defines the classic *APOE* haplotypes ($\epsilon_2$, $\epsilon_3$, $\epsilon_4$) [17]. The detailed results for the 11 SNPs are provided in Table 2. The positive relationship between the statistical significance obtained from the proposed indirect test and the interaction test is further illustrated in the scatterplot shown in Fig 4(c).

To confirm that accounting for interaction with rs7412 can indeed explain part of the non-additive (i.e., dominance) effects flagged earlier by the proposed indirect test, we re-applied the test to the 65 selected SNPs, both with and without including the interaction term with rs7412. When the interaction is not considered, we found 10 SNPs with p-values from the

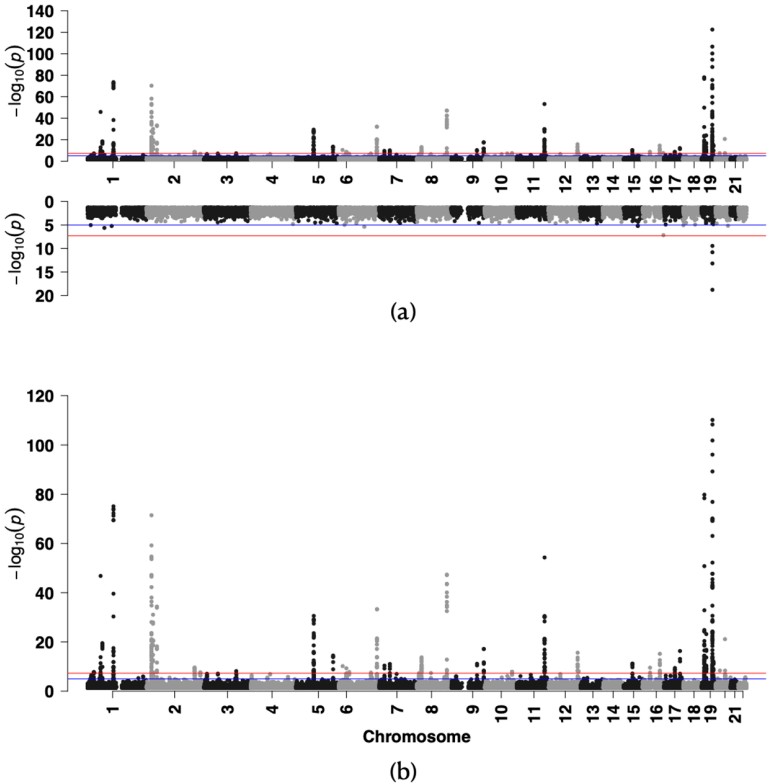

**Fig 3. (a): Miami plot of the GWAS result using the proposed 2-df joint test (upper) and the non-additive (dominance) test (bottom).** The red line denotes the genome-wide significance level of 5e-8. (b): Manhattan plot of the GWAS result using the traditional additive test, where the dominance term is not included in the model. For visualization purposes, only SNPs with p-values less than 0.05 were shown in the plot: 39,473 SNPs for the joint test, 44,881 for the dominance test, and 29,640 for the traditional additive test.

non-additive test less than 0.05, and 3 SNPs with p-values less than 5e-8. After accounting for the interaction effect, the (-log10) p-values, and the magnitudes of the estimated non-additive effects, are shrunk for 8 of the 10 SNPs, as summarized in Fig 5. In particular, none of the 10 SNPs has a genome-wide significant p-value of the non-additive test after their interactions with rs7412 are accounted for.

## Discussion

Using heteroskedasticity to indirectly test for a latent interaction is well-established in the analysis of quantitative traits, and has led to many scientific insights over the human genome. However, none of the existing approaches of indirect testing could be applied when the trait of interest is binary. In this paper, we (i) derive, for the first time, an indirect test for binary traits, and in doing so, (ii) offer a practical interpretation for non-additive effects identified in binary trait GWAS. The proposed method requires only the addition of a non-additive (dominance) term to the conventional additive regression model and can be implemented directly using the PLINK2 GWAS software [3]. For other commonly used GWAS software, integrating the proposed methodology may involve some additional coding, but the overall implementation should be manageable. We have applied this method both in the simulation studies

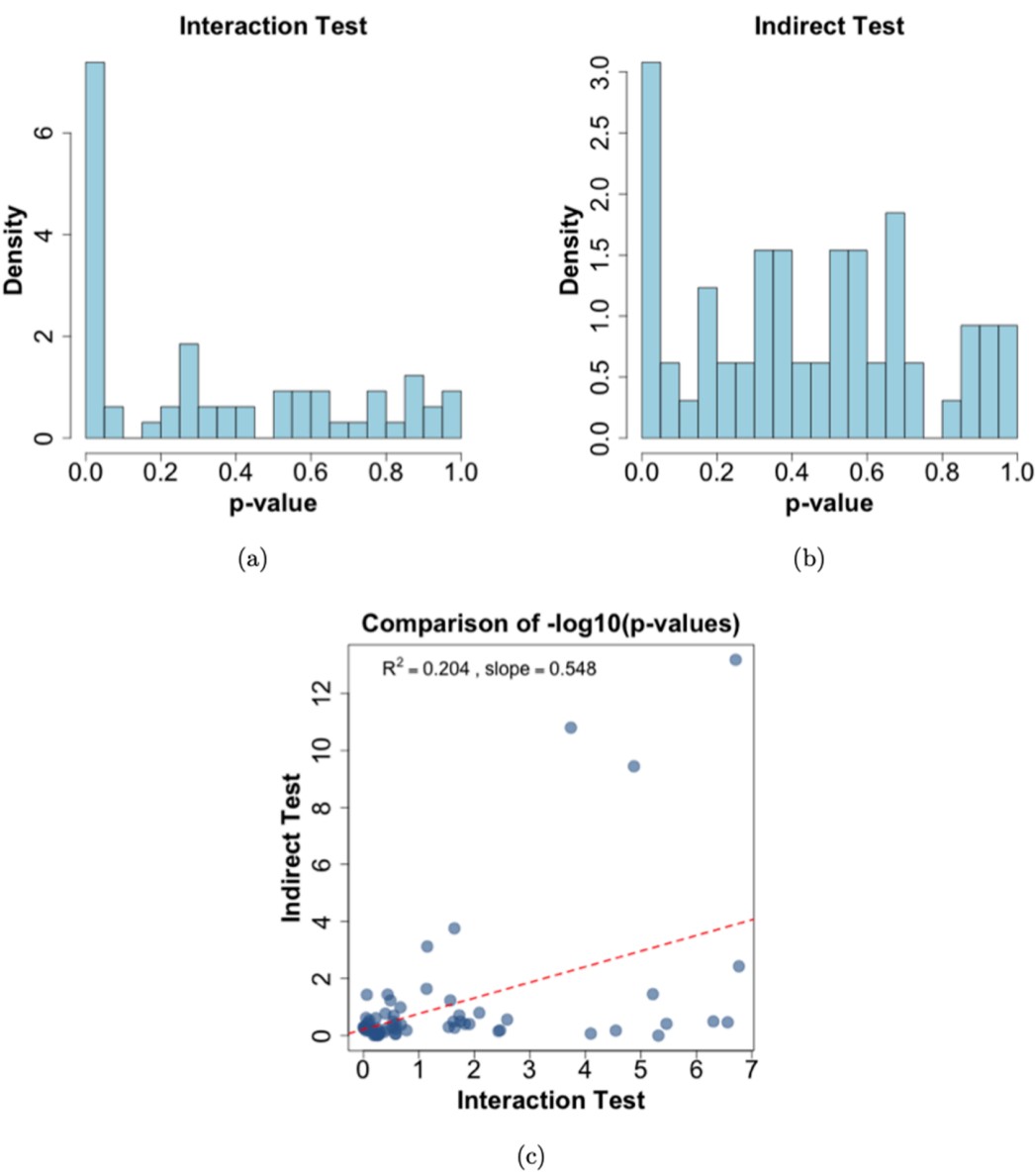

**Fig 4. (a–b): Histograms of p-values for the 65 selected SNPs, testing either the interaction effect with rs7412 (a) or the non-additive effect (b).** (c): Scatterplot of $(-\log_{10})$ p-values from the indirect (non-additive) test versus the interaction test. The red dashed line shows the fitted regression line, with a slope of 0.548 and $R^2$ of 0.204.

and the analysis of the binary trait self-reported high cholesterol in the UKB data, and found promising SNPs with supporting evidence from the existing literature.

It has been suggested in the literature that non-additive genetic effects do not explain as much variability as the additive effects in most human traits [19], supported by the weak dominance signals identified from the dominance GWAS scan. Furthermore, [10] has shown for binary traits that the non-additive signals are breaking down more rapidly as the linkage disequilibrium breaks down, which has been viewed as another reason to prefer the use of the additive-only model and to ignore the non-additive component in the analysis. Although non-additive signals may not be as prevalent as the additive signals across the genome in a

**Table 2. Summary of characteristics of the 11 SNPs identified through the interaction analysis; including minor allele frequencies, linkage disequilibrium measures (D' and $R^2$), distance (in BP) to rs7412, and association test p-values (indirect and interaction) along with estimated regression coefficients ($\hat{\beta}_D$ and $\hat{\beta}_{GE}$). As a comparison, the D' to rs429358 is also shown in the parenthesis.**

| Rank | SNP | MAF | D' | $R^2$ | Distance | P:Indirect | P:Interaction | $\hat{\beta}_D$ | $\hat{\beta}_{GE}$ |
|---|---|---|---|---|---|---|---|---|---|
| 1 | rs7254892 | 0.031 | 0.931 (1.000) | 0.413 | −22483 | 3.723e-03 | 1.713e-07 | −0.0184 | 0.04826 |
| 2 | rs141622900 | 0.045 | 0.953 (1.000) | 0.636 | 14713 | 6.685e-14 | 1.960e-07 | −0.0474 | 0.06425 |
| 3 | rs34954997 | 0.219 | 1.000 (0.984) | 0.239 | 5559 | 3.402e-01 | 2.733e-07 | −0.0060 | 0.05764 |
| 4 | rs483082 | 0.219 | 1.000 (0.984) | 0.239 | 4099 | 3.166e-01 | 4.940e-07 | −0.0063 | 0.05684 |
| 5 | rs405509 | 0.484 | 1.000 (0.602) | 0.063 | −3243 | 3.849e-01 | 3.467e-06 | −0.0052 | −0.04932 |
| 6 | rs440446 | 0.363 | 1.000 (0.982) | 0.038 | −2912 | 9.960e-01 | 4.849e-06 | 0.0000 | −0.04909 |
| 7 | rs75627662 | 0.186 | 1.000 (0.748) | 0.293 | 1497 | 3.514e-02 | 6.094e-06 | −0.0134 | 0.05253 |
| 8 | rs72654473 | 0.093 | 1.000 (0.182) | 0.648 | 2320 | 3.613e-10 | 1.337e-05 | −0.0425 | 0.06595 |
| 9 | rs439401 | 0.381 | 1.000 (0.983) | 0.041 | 2372 | 6.640e-01 | 2.835e-05 | 0.0026 | −0.04502 |
| 10 | rs584007 | 0.378 | 1.000 (0.983) | 0.041 | 4399 | 8.491e-01 | 8.011e-05 | −0.0012 | −0.04235 |
| 11 | rs445925 | 0.094 | 1.000 (0.190) | 0.641 | 3561 | 1.593e-11 | 1.825e-04 | −0.0419 | 0.06027 |

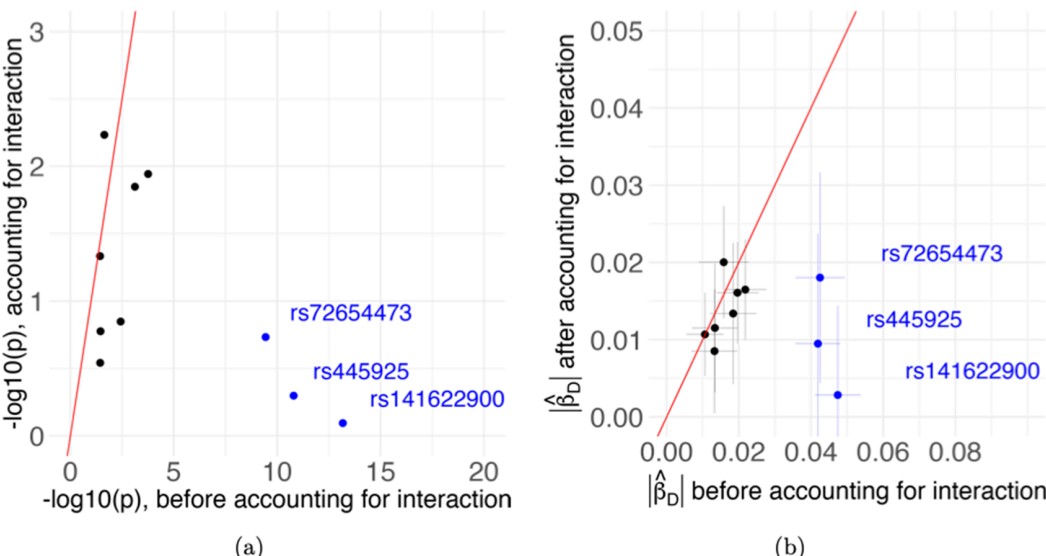

(a)  (b)

**Fig 5. The (-log10) p-values of the non-additive test (a) and the absolute values of estimated non-additive effects (b) before (x-axis) and after (y-axis) accounting for the interaction with rs7412, for the ten SNPs with p-values of non-additive test less than 0.05.** The three SNPs with genome-wide significant p-values of the non-additive test before accounting the interaction are highlighted in blue. The red line is the line of $y = x$. The radius of each cross in (b) denotes the standard error of the non-additive effect estimate.

marginal scan, our works suggests that joint testing of the non-additive and the additive effect may uncover many SNPs that could not be identified by the traditional additive test alone. At the same time, our study illustrates that for binary traits, the additive and non-additive effects of SNPs cannot be easily interpreted separately. Therefore, these two effects should be jointly tested and interpreted together as the genetic effect in binary trait GWAS.

Our approach has a similar nature to the approach of [22] for quantitative traits, where [22] accounts for latent interactions by jointly testing the genetic effects in the location and the scale of the quantitative trait, and our approach accounts for latent interactions by jointly testing the additive and non-additive genetic effects for the binary trait. In [22], it is emphasized that we cannot conclude whether the heteroskedasticity is caused by the SNP itself, or by a latent interaction. Similarly for our approach, a significant non-additive effect

could be either due to the biological mechanism of the SNP itself, or its interaction with latent variables. These joint tests are valuable for identifying SNPs for more detailed interaction analysis, but their results should not be over-interpreted.

## Supporting information

**S1 Table. Different parameterization of the saturated model: The parameters parameterization of $\gamma_0, \gamma_1, \gamma_2$ in terms of $\beta_0, \beta_G, \beta_{GE}$ assumes that the original model is probit.** However, as explained in the paper, there will be no model-misspecification issue even when the original model is not probit. The notation $p$ denotes the minor allele frequency and $q = 1 - p$. (PDF)

**S2 Table. Summary of characteristics of all the 65 SNPs studied in the interaction analysis; including minor allele frequencies, linkage disequilibrium measures (D' and $R^2$), distance (in BP) to rs7412, and association test p-values (indirect and interaction) along with estimated regression coefficients ($\hat{\beta}_D$ and $\hat{\beta}_{GE}$).** (PDF)

**S1 Fig. (A-B) show contours of $\gamma_D$ and (C-D) show heat-maps of the non-additive proportion of genetic variation $R_D^2$, at different $\beta_G$ and $\beta_{GE}$.** The parameter $\beta_E = 0.5$ and $E \sim N(0,1)$. The MAF is set to $p = 0.3$ in (C-D). The underlying model is assumed to be probit. The prevalence of the binary trait $Y$ is 0.1 on the left column and 0.3 on the right column. (PDF)

**S2 Fig. The histograms for the p-values of the proposed indirect test when $\beta_{GE} = 0$, at different settings of the MAF and $\beta_E$.** The individuals ($n = 100,000$) are simulated from a logistic regression model with $\beta_G = 0.5$ and $\beta_0 = -1$. The latent variable $E$ follows $N(0,1)$ independent of the SNP. (PDF)

**S3 Fig. The histograms for the p-values of the proposed joint test when $\beta_{GE} = \beta_G = 0$, at different settings of the MAF and $\beta_E$.** The individuals ($n = 100,000$) are simulated from a logistic regression model with $\beta_0 = -1$. The latent variable $E$ follows $N(0,1)$ independent of the SNP. (PDF)

**S4 Fig. The histograms (a-b) and QQ-plots (c-d) for the GWAS p-values (indirect in the left, joint in the right), for the European population.** The binary trait (high cholesterol) has been permuted before the GWAS. (PDF)

**S5 Fig. Power of the proposed tests: The power for the proposed indirect test of $\beta_{GE}$ based on the non-additive effect $\beta_D$ is shown in the first row, and the power for the proposed joint test of $\beta_{GE}$ and $\beta_G$ is shown in the second row.** The size of $\beta_E$ is respectively set to $-0.5$ (left), 0 (center) and 0.5 (right). The sample size is $n = 30,000$. (PDF)

**S6 Fig. Power of the proposed tests: The power for the proposed non-additive based on $\beta_D$ is shown in the first row, and the power for the proposed joint test of $\beta_{GE}$ and $\beta_G$ is shown**

**in the second row.** The size of $\beta_E$ is respectively set to –0.5 (left), 0 (center) and 0.5 (right). The sample size is $n = 300{,}000$, which is comparable to the size of modern Biobanks. (PDF)

**S7 Fig. Power of the proposed tests: The power for the proposed indirect test of $\beta_{GE}$ based on the non-additive effect $\beta_D$ is shown in the first row, and the power for the proposed joint test of $\beta_{GE}$ and $\beta_G$ is shown in the second row.** The size of $\beta_E$ is respectively set to –0.5 (left), 0 (center) and 0.5 (right). The sample size is $n = 800{,}000$. (PDF)

**S8 Fig. The histograms (a-b) and QQ-plots (c-d) for the GWAS p-values (indirect in the left, joint in the right), for the European population.** (PDF)

## Acknowledgments

Ziang Zhang was a trainee of the CANSSI-ONTARIO STAGE (Strategic Training for Advanced Genetic Epidemiology) training program at the University of Toronto.

## Author contributions

**Conceptualization:** Ziang Zhang, Lei Sun.

**Formal analysis:** Ziang Zhang, Jerald F Lawless, Andrew D Paterson.

**Funding acquisition:** Lei Sun.

**Investigation:** Ziang Zhang, Andrew D Paterson, Lei Sun.

**Methodology:** Ziang Zhang, Jerald F Lawless, Lei Sun.

**Supervision:** Jerald F Lawless, Andrew D Paterson, Lei Sun.

**Validation:** Ziang Zhang, Jerald F Lawless, Andrew D Paterson, Lei Sun.

**Visualization:** Ziang Zhang, Andrew D Paterson, Lei Sun.

**Writing – original draft:** Ziang Zhang.

**Writing – review & editing:** Ziang Zhang, Jerald F Lawless, Andrew D Paterson, Lei Sun.

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
