## [Decision Letter · Decision Letter 0]

4 Apr 2025

PGENETICS-D-24-01489

Detecting latent gene-environment interaction when analyzing binary traits

PLOS Genetics

Dear Dr. Zhang,

Thank you for submitting your manuscript to PLOS Genetics. After careful consideration, we feel that it has merit but does not fully meet PLOS Genetics's publication criteria as it currently stands. Therefore, we invite you to submit a revised version of the manuscript that addresses the points raised during the review process.

Please submit your revised manuscript within 60 days Jun 03 2025 11:59PM. If you will need more time than this to complete your revisions, please reply to this message or contact the journal office at plosgenetics@plos.org. Please include the following items when submitting your revised manuscript:

We look forward to receiving your revised manuscript.

Kind regards,

Heather J Cordell

Academic Editor

PLOS Genetics

Michael Epstein

Section Editor

PLOS Genetics

Aimée Dudley

Editor-in-Chief

PLOS Genetics

Anne Goriely

Editor-in-Chief

PLOS Genetics

**Journal Requirements:**

https://journals.plos.org/plosgenetics/s/submission-guidelines#loc-parts-of-a-submission

5) Please ensure that the funders and grant numbers match between the Financial Disclosure field and the Funding Information tab in your submission form. Note that the funders must be provided in the same order in both places as well.

**Reviewers' comments:**

Reviewer's Responses to Questions

**Comments to the Authors:**

Reviewer #1: There is a large body of literature showing that when a genetic variants is involved in an interaction effect on a given continuous outcome, it is expected to be associated with the variance of that outcome. Based on this observation, various approaches have proposed to pre-select candidate for interaction effect testing based on variance or other scale metrics. This paper proposes an equivalent of that approach for the case of binary outcomes. I actually briefly looked at this question a few years ago but didn’t found any solution, so I would like to commend the author for figuring out the proposed approach. Although the path toward the solution is not trivial, the final model is very simple and only consists in testing the significance of a dominance term (on top of the standard additive one). I ran a few toy simulations and qualitatively confirmed the results from the authors: that is, the presence of a dominance effect when a GxE interaction effect is added to the generative model. Overall, the manuscript reads well and I do not have any major concern. My minor comments are:

1. As mentioned above, the proposed solution is very simple in practice: adding a dominance term on top of the additive one. It took me multiple reading and running simulation to make sur I got this right. This should be much clearly stated along the manuscript (abstract, intro and conclusion).

2. I found Figure 1 and 2 pretty hard to read. I understand that they show the big picture, but given the shape of the curves I am not sure what are the take-home messages. There are simpler plots that would better convey the message found in the main text. For example, for Fig.2, the author state that “the proposed non-additive and joint tests tend to have higher power when betaGE and betaE have opposite signs”. A plot showing the power of the tests on the Y axis, and betaGE on X axis would be easier to read. The authors may consider moving those two figures to supp and replace them with plots illustrating the specific characteristics they would like to highlight.

3. Comparing Figure S8 and Figure 3a, there is something that I do not understand. The p-values from the joint test (Figure 3a) appear to be substantially lower than for the marginal (Figure S8) for most of the peaks. For example on chromosome 1, the lowest P for the additive model is ~1e-40, when in comparison it is ~1e-80 for the joint test. The min(P) for the dominance test on that chromosome is 1e-5. From my experience, the joint test is expected to be fairly close to the marginal one, especially if the added term (here the dominance) has a modest effect. I ran multiple simulations but couldn’t produce a single scenario where the difference was so large. Some explanation are welcome.

4. BTW. The authors may consider moving Figure S8 along the other Manhattan in Figure 3 to allow for a direct comparison of the proposed test with the standard additive logistic model.

5. In Figure 3: it seems that plotting the interaction test against the indirect test against each other (-log10(Pinteraction) = f(-log10(Pindirect))) would be more informative if the goal is to highlight the link between the two approaches.

6. There are 10 SNPs in Fig 4, but 11 discussed in the earlier part of the results. What happen to the last one?

7. Introduction: “However, a direct, exhaustive GxG search may be undesirable […] multiple hypothesis testing”. May be worth mentioning the computational burden as well!

8. Introduction: Typo “have also been reported [by] Singhal et al. (2023).”

9. There are parentheses missing in second and third lines of equation (8)

10. Please provide a reference for the “Fisher orthogonal re-parametrization”. I personally never heard about it before.

Reviewer #2: Uploaded as an attachment

**Have all data underlying the figures and results presented in the manuscript been provided?**

Reviewer #1: Yes

Reviewer #2: Yes

PLOS authors have the option to publish the peer review history of their article (what does this mean?). If published, this will include your full peer review and any attached files.

Reviewer #1: No

Reviewer #2: No

**Figure resubmission:**
---

## [Decision Letter · Decision Letter 1]

20 Jul 2025

PGENETICS-D-24-01489R1

Detecting latent interaction effects when analyzing binary traits

PLOS Genetics

Dear Dr. Zhang,

Thank you for submitting your manuscript to PLOS Genetics. After careful consideration, we still feel that it has merit but does not fully meet PLOS Genetics's publication criteria as it currently stands. Therefore, we invite you to submit a revised version of the manuscript that addresses the remaining points raised during the review process.

Please submit your revised manuscript within 30 days Aug 19 2025 11:59PM. If you will need more time than this to complete your revisions, please reply to this message or contact the journal office at plosgenetics@plos.org. Please include the following items when submitting your revised manuscript:

We look forward to receiving your revised manuscript.

Kind regards,

Heather J Cordell

Academic Editor

PLOS Genetics

Michael Epstein

Section Editor

PLOS Genetics

Aimée Dudley

Editor-in-Chief

PLOS Genetics

Anne Goriely

Editor-in-Chief

PLOS Genetics

**Reviewers' comments:**

Reviewer's Responses to Questions

Reviewer #1: I thank the authors for their clear responses to all my comments. I have one important additional comments related to point #3 ("Comparing Figure S8 and Figure 3a,...")

First, the simulations from Figure R1 and the explanation provided are fully in line with my understanding of the joint test behavior. Still, one point needs clarification. On line 347-348, the authors state “For comparison, the Manhattan plot from the traditional additive GWAS is also shown in Figure [3](b).” It looks like the authors are referring to the 1df test of beta_A obtain from a PLINK2 run where both G_A and G_D have been modelled (and the R script provided on git appears to agree with this). If so, this is unfair (for the reasons provided by the authors in their response) and the “traditional additive GWAS” should refer to marginal additive effect of the variants where the dominance effect is not modelled at all (i.e. the model Y = G + covariates, where G is coded as 0,1,2). Assuming I am correct, the authors should update Figure 3b and replace it with a standard additive GWAS.

**Have all data underlying the figures and results presented in the manuscript been provided?**

Reviewer #1: Yes

PLOS authors have the option to publish the peer review history of their article (what does this mean?). If published, this will include your full peer review and any attached files.

Reviewer #1: No

**Figure resubmission:**
---

## [Editor Report · Decision Letter 2]

24 Jul 2025

Dear Dr Zhang,

We are pleased to inform you that your manuscript entitled "Detecting latent interaction effects when analyzing binary traits" has been editorially accepted for publication in PLOS Genetics. Congratulations!

In the meantime, please log into Editorial Manager at https://www.editorialmanager.com/pgenetics/ click the "Update My Information" link at the top of the page, and update your user information to ensure an efficient production and billing process. Note that PLOS requires an ORCID iD for all corresponding authors. Therefore, please ensure that you have an ORCID iD and that it is validated in Editorial Manager. To do this, go to ‘Update my Information’ (in the upper left-hand corner of the main menu), and click on the Fetch/Validate link next to the ORCID field.  This will take you to the ORCID site and allow you to create a new iD or authenticate a pre-existing iD in Editorial Manager.

Yours sincerely,

Heather J Cordell

Academic Editor

PLOS Genetics

Michael Epstein

Section Editor

PLOS Genetics

Aimée Dudley

Editor-in-Chief

PLOS Genetics

Anne Goriely

Editor-in-Chief

PLOS Genetics

Comments from the reviewers (if applicable):

**Data Deposition**

http://datadryad.org/submit?journalID=pgenetics&manu=PGENETICS-D-24-01489R2

**Press Queries**

---

## [Editor Report · Acceptance letter]

PGENETICS-D-24-01489R2

Detecting latent interaction effects when analyzing binary traits

Dear Dr Zhang,

We are pleased to inform you that your manuscript entitled "Detecting latent interaction effects when analyzing binary traits" has been formally accepted for publication in PLOS Genetics! Your manuscript is now with our production department and you will be notified of the publication date in due course.

With kind regards,

Anita Estes

PLOS Genetics

On behalf of:
